# Differences in Workplace Risk Perception between Foreign-Born and First-Generation Mexican American Construction Workers

**DOI:** 10.3390/ijerph18115652

**Published:** 2021-05-25

**Authors:** Gabriel Ibarra-Mejía, Karla Gabriela Gómez-Bull, María Marisela Vargas-Salgado

**Affiliations:** 1Public Health Sciences Department, The University of Texas at El Paso, 500 West University Avenue, El Paso, TX 79968, USA; gabmejia@utep.edu; 2Industrial and Manufacturing Engineering Department, Universidad Autonoma de Ciudad Juárez, Av. Plutarco Elías Calles #1210, Ciudad Juarez, Chihuahua 32310, Mexico; 3Administrative Sciences Department, Universidad Autonoma de Ciudad Juárez, Av. Plutarco Elías Calles #1210, Ciudad Juarez, Chihuahua 32310, Mexico; maria.vargas@uacj.mx

**Keywords:** risk perception, construction industry, foreign workers

## Abstract

Risk perception is used to quantify risks in the industry and is influenced by different socio-demographic variables. This work aims to determine significant differences in the risk perception between Mexican American migrants and first-generation Mexican American construction workers. This study used a sample of 112 construction workers. A guided questionnaire was applied to collect socio-demographic information. For workplace risk behaviors, we used a 21-item questionnaire adapted from the previous instrument. Each question asked the participant’s perception of the frequency with which they carried out risky activities during routine work activities and the severity of the possible injuries, using a five-level Likert scale. Then, an inferential analysis was carried out using analysis of variance (ANOVA). The main results highlight that time of residence in the United States had a significant influence (*p* = 0.012) on risk perception in the surveyed construction workers. On the other hand, the age and time they have been working for the organization did not significantly influence risk perception. Finally, risk perception can vary in construction workers according to different variables. It is essential to investigate the factors that influence it, to prevent risky behaviors that can lead to accidents.

## 1. Introduction

Within the construction industry research community, there is agreement that construction work is, first, one of the largest industries in the United States, and second, one of the most dangerous and indicative of a deteriorating situation of the construction industry [1]. Accidents in this field are associated with the site layout, materials, tools and equipment, and trade workforces that make up a volatile site environment [2]. According to United States data, workers in the construction industry account for about six to eight percent of all workers, and injuries and deaths are not uncommon. Although the construction industry sector has had significant technological and procedural advances, these have not always positively affected risk and accident prevention [3]. Nowadays, construction workers must shoulder a high degree of risk [4,5]. Studies examine the factors underlying why construction is the industry with the highest rate of occupational injuries continues to date.

International and local construction firms helped the economy of the USA, providing at the same time employment opportunities for many [6]. Construction will continue adding jobs in the goods-producing sectors. According to the Bureau of Labor Statistics (BLS), by 2024, construction industry jobs are projected to reach more than 6.9 million. It will also be the fourth fastest-growing industry sector, which, regarding output, is the fastest of all goods-producing industries and growing faster than the economy as a whole [7]. Furthermore, by 2024, construction industry jobs are projected to reach more than 6.9 million. This industry involves numerous factors that represent a potential danger to workers [8]. The pure labor-intensive nature of construction work and its higher occupational-related injuries pose a significant management challenge for the industry.

In the United States, there has been an increase in the foreign-born population, even though the general population growth rates have decreased steadily since the 2007 great recession. The role of immigrant workers in the USA construction industry has become of great significance. An increasing number of foreign-born individuals from the Hispanic population are joining the construction industry workforce. An estimated quarter of the total industry workforce comprises foreign-born workers, out of which 84% come from Latin-American countries [9]. However, disproportionally high rates of immigrants work in high-risk occupations, such as in the construction industry.

Interestingly, considering an estimated working lifespan between 20 and 65 years of age, construction workers in the United States have an overall 75% probability of undergoing a disabling injury [10]. However, for Hispanic construction workers, this percentage increases to 90%, and their risk of death is on fifth higher than the overall death risk [10]. Furthermore, disparately high occupationally related death and injury rates exist between foreign-born Hispanic construction workers and native Hispanic construction workers. Foreign-born Hispanic construction workers are the most vulnerable to fatality, and injuries in foreign-born workers constitute 74% of deaths among Hispanic construction workers, making them the most vulnerable population group for occupationally related deaths [11].

According to recent statistics from the Bureau of Labor Statistics, accidents in the construction industry increased by 5% since 2007 in the United States of America, and 1061 cases of fatal occupational injuries were registered [12]. Currently, there are no specific statistics from the BLS on Mexican American industries.

Risk perception can be defined as a process used to evaluate the amount of risk present in a certain situation; it reflects uncertainty, and it is different from person to person [13]. While individuals perceive risks and have concerns, it is the culture that provides socially constructed myths about nature systems of beliefs reshaped and analyzed by persons, becoming part of their worldview and influencing their interpretation of natural phenomena [14]. However, progress has been slow in explaining gender differences in perceived risk, and few studies have examined how differences relate to other characteristics of individuals, such as race [15,16] affect perceived risk. Humans learn to believe that the standards, principles, perspectives, and explanations that we acquire from our culture are the way to look at the world [17]. Pre-established cultural beliefs help people make sense of risk, and notions of risk are therefore not individualistic but instead shared within a community [18]. Theoretically, a worker’s internal factors, such as attitude, perception, and efficacy, play a vital role in safety performance [1,19,20,21].

In addition, besides the construction industry, risk perception has been addressed in different contexts such as manufacturing [22], tourism [23], driving behavior [24], and disaster risk reduction [25], among others. Additionally, construction workers’ risk-taking behavior is negatively influenced by risk perception and job environments [26], which may lead to construction employees having a high rate of work-related injuries and a low rate of personal protective equipment (PPE) use [27]. However, little is known about why different people have different risk tolerances, even when confronting the same situation [1]. Specifically, among Hispanic-Latino construction workers, several factors have been associated with increased workplace risk. These risk factors are: (a) a lack of formal education or little training to perform the job; (b) a lack of job training or provision of safety equipment by companies to their employees; (c) language and literacy barriers; (d) negligence related to observing safety measures; and, (e) the very need for a job, which leads the worker to expose him or herself to certain risks for fear of losing it [28]. All of these may relate with specific cultural aspects that place the Hispanic-Latino worker at a disadvantage, therefore, to a higher risk of work-related accidents and injuries. Therefore, undermining the significance of the role of this population group and its socio-demographic characteristics in the USA construction industry may affect the design, implementation, and effectiveness of injury and death preventative efforts.

As stated before, a worker’s internal factors such as attitude, perception, and efficacy play a vital role in safety performance. Risk perception has important effects not only on safety behaviors, but also on the decisions of workers for the use or non-use of personal protective equipment [29] (12), motivation [21], and the effective identification of hazards [30]. Based on the assumption that risk perception is a determinant to a person’s health behaviors [31,32], we studied the differences in workplace risk perception and risk behavior between two groups of construction workers of Mexican origin—one group of Mexican American migrants (foreign-born) and first-generation Mexican Americans. The study was conducted in the Paso Del Norte region, located on the USA–Mexico border, whose unique setting provides opportunities to study such phenomena. Our overarching goal is to contribute to the design of culturally appropriate interventions, which can ultimately lead to a decreased likelihood of work-related accidents and injuries in this industry and for this population group. This project was funded by the “Programa de Investigación en Migración y Salud—PIMSA” From the University of California–Berkeley. A requisite for the research funding by this agency is that it contemplates health and occupational issues in migrant populations from Latin America in the United States. Therefore, the focus must be kept within the population of interest.

## 2. Materials and Methods

### 2.1. Participants

Participants were recruited at different construction worksites throughout the El Paso (Texas) and Dona Ana (New Mexico) Counties. Eligible candidates for enrollment were self-reported Mexican migrants, and first-generation Mexican American workers from construction contractors and companies, and non-governmental organizations (NGOs) associated with construction work in the Paso Del Norte region. Participants involved in the study were either male or female construction workers. All participants were 18 years of age and older, able-bodied, with at least a six month history of uninterrupted construction work history, and currently living in the Paso del Norte region. Workers who did not meet inclusion criteria or had a known self-reported history of musculoskeletal, neurological, and metabolic chronic disease were not included in the study. Afterward, eligible participants were invited and selected at convenience until the sample size quota was met. The study was approved by the University of Texas at El Paso’s Institutional Review Board (FWA No: 00001224). All eligible participants received an explanation of the study’s purpose and procedures and signed informed consent.

### 2.2. Methods

A guided questionnaire was employed to collect demographic, socio-economic, and cultural information from participants. Workplace risk behaviors were explored using a 12-item questionnaire adapted from a previous instrument from Xia et al. [33] to assess risk perception. The first six items correspond to the perceived probability, and the next six refer to perceived severity. From here risk perception was calculated by multiplying both values. We previously validated this instrument, where we obtained a Cronbach’s α greater than 0.7 (perceived probability α = 0.798, perceived severity α = 0.797, risk perception α = 0.842), indicating that the instrument is reliable. Each question asked the participants’ perception of the frequency with which they carried out risky activities during routine work activities. The level of risk perception was evaluated based on a subjective evaluation of probability and the severity of the consequence during six of the most frequent routine tasks performed by construction workers in the United States of America according to reported data from the USA Bureau of Labor Statistics; these six tasks are the most representative activities of construction [34], as shown in Table 1. The level of perception for each item was scored using a five-level Likert scale.

### 2.3. Procedure

A list of major construction companies in the Paso Del Norte border region was used as a source to identify potential participants. Data collection was carried out during the period of December 2017 and January 2018. During that period, there were 2570 construction workers in El Paso, TX [35]. Therefore, the sample size calculated with a 95% confidence level was 93. Direct contact was established by telephone or electronic mail with a company’s project representative. Once companies agreed to allow access to their worker population, initial interviews were conducted to screen eligible participants. After confirming for selection criteria and upon accepting to participate, a research assistant explained the purpose of the study and asked to read the informed consent. Once consent was obtained, each participant completed a self-administered questionnaire, collecting demographic, socio-economic, and cultural information.

### 2.4. Data Analysis

After completing the questionnaire, participants were assigned to one of two groups: (1) foreign-born Mexican migrants and (2) first-generation Mexican immigrants. Descriptive analyses were conducted for both groups through the statistical analysis software SPSS for Windows version 25.0 (IBM Corporation, Armonk, NY, USA). KMO and Bartlett’s tests were obtained to analyze the adequacy of the data to a factor analysis model.

ANOVA tests were performed using SPSS statistical software to identify significant differences between the place of birth, residence time in the USA, and time worked in the organization, intending to determine if these factors influence risk perception. Finally, a Tukey test was developed to find differences between groups of the study regarding the variables with significant differences in risk perception.

## 3. Results

Table 2 shows the demographic characteristics of the study sample. One hundred sixteen construction workers participated in the study, of which four were excluded for missing answers. The final sample consisted of 112 subjects. The availability of workers influenced the study’s sample size during their work shift. Additionally, according to the Bureau of Labor Statistics, there were 2570 construction workers in El Paso, TX, during the period in which data collection began. Therefore, sample size estimation (with a 95% confidence level) yielded 93 participants. These parameters are considered adequate for exploratory research. There was only one female participated in the sample, which is characteristic of the region where very few women work for construction companies. This finding is consistent with the reports from the National Association of Women in Construction (NAWIC) [36], which states that according to the BLS [37], only 1.5% of the entire United States construction workforce is comprised of women. Concerning nationality, 56.28% were from Mexico, while 43.75% claimed to be from the United States of America. Most of the sample (75.89%) had more than 12 months working in the construction industry. Regarding age, 16.96% were between 26 and 30 years old, 16.07% between 41 and 45 years old, and 13.39% between 20 and 25 years. Most of them had more than six years of residence in the United States of America (81.25%). Concerning time worked in the organization, 34.82% had more than six years working for it.

Table 3 contains statistics about the probability and severity of injuries and risk perception for each routine construction task. Concerning perception, workers perceived high probabilities of injuries when helping fellow workers (4.0982) and using work tools (3.9375). On the other hand, subjects perceived less probability of injuries when mounting or dismounting structures and scaffoldings (2.5804). Results of severity coincide with probability since workers perceived high severity when helping fellow workers (3.875) and using work tools (3.8304)—perceiving less severity when mounting or dismounting structures and scaffoldings. According to the results, risk perception was higher when construction workers helped other colleagues (16.7) and when they used work tools (15.9).

Table 4 contains the results of the factorial analysis at the construct level for risk perception, with Varimax rotation, with the number of factors as an extraction criterion (using 1) with regression as a scoring method. The Kaiser–Meyer–Olkin (KMO) value was 0.784, showing that it has a regular or medium factorial adjustment [34]. On the other hand, Bartlett’s Sphericity test indicator, a statistical test for the presence of correlations between the variables, was 432.086, with 15 degrees of freedom [38].

Results of analysis of variance for risk perception are shown in Table 5. We found that the age and time participants have been working for the company did not significantly influence risk perception (0.890 and 0.136, respectively). On the contrary, time of residence in the United States significantly influenced risk perception among construction workers (0.12). This finding means that the number of years they have resided in that country influenced how they perceived the activities carried out in their work, with differences in the probability and severity of an accident occurring while performing their tasks.

A Tukey test was performed for the variable time of residence in the United States, which obtained significant differences between the groups. The results are shown in Table 6, where between groups 3 and 4, there are significant differences in the risk perception of construction workers. In other words, workers who had been living between four and six years in the United States perceived the risk differently from those who had more than six years living in that country.

## 4. Discussion

This work aimed to determine differences in risk perception in construction workers as a result of socio-demographic variables such as age, time of residence in the United States, and time working for the organization. The sample of construction workers for this study showed that age and time working for the company did not influence risk perception. These results agree with Chan et al. [15].

The results found in this work coincide with those found by Lopez et al. [39] and Ricci et al. [40], who detected differences in the risk perception of construction workers from different nationalities. These differences reinforce our hypothesis that culture can significantly affect the way construction workers perceive risk situations. Nevertheless, still, little is known about the effect of cultural beliefs. On the contrary, other authors have not found differences in risk perception, establishing that culture did not influence the obtained results. However, it is essential to highlight that more research is needed to clarify differences in the risk perception of construction workers. It is relevant to analyze why workers with different cultural characteristics perceive the same risk situations differently [5].

The results found in the analysis of variance showed no significant differences in the risk perception of construction workers concerning age. These findings agree with Oah et al. [22]. On the other hand, our results differ from the findings of Ellaban et al. [41], Trillo-Cabello et al. [5], Chaswa et al. [42], and Forcael et al. [3], who found that the risk perception was different depending on the age of the participants, for which they mention that this demographic variable is a factor that can influence such perceptions.

In this study, the time workers have been working in the organization did not significantly influence risk perception. This finding agrees with Rodríguez-Garzón et al. [43], who also analyzed the differences in risk perception based on demographic variables, finding that the time working in that job position does not impact the risk perception.

This research also has some limitations. First, this work used a cross-sectional design investigation with a non-probabilistic sample due to the availability of the participants. The CoWoRP scale from Man et al. [44] was not used because our work was the first study in a set of investigations. Moreover, our data collection was carried out from December 2017 to January 2018, before the CoWoRP scale was published. This study was limited to the analysis of the cognitive component of risk. Future studies should address risk perception from its cognitive and affective components using another instrument such as the CoWoRP scale to compare the results. It is important to increase age ranges in the following studies.

It is recommended to explore other cultural variables that could influence the risk perception of construction workers and inquire about the accident history of the participants. It would also be interesting to analyze whether this same phenomenon occurs in workers from other work contexts who have lived in the United States for a longer time.

In addition, some organizational factors can also have impact on the risk perception, such as safety climate, leadership levels, workloads, motivation of workers, and knowledge and safety training; it is important to consider the effect of these variables so as to expand information about risk perception.

Additionally, personal traits, attitudes, and personality, as well as history of accidents at work or experiences with fatalities, must be included in following works. It is considered essential to determine the factors from which workers may perceive risk differently, as it can serve as a basis for future interventions that help manage risks in the workplace and prevent risk behaviors.

## 5. Conclusions

Construction is a risky industry and understanding the sources of the risks is essential to implementing appropriate risk prevention and mitigation methods [13]. Despite all of these, there have been very few studies on the health-related accident status of migrant workers [45].

We found that time of residence in the United States had a significant effect on risk perception. This finding brings more clarity that nationality or cultural characteristics of subjects are factors that can interfere with the risk perception. It can provide some clarity on the reason why workers can face the same risk situation but may make different decisions regarding it. Despite this, the study sample perceived that among the activities commonly carried out in the construction industry, those with the highest risk were helping other colleagues and when using work tools.

However, the main results of this work highlight that time of residence in the United States of America had a significant influence on risk perception. Our results identified differences associated with the length of residence. This review should provide a basis for conducting research focused on work-related risk perception among immigrant workers, which will influence accident rate reduction in this particular worker population.

## Figures and Tables

**Table 1 ijerph-18-05652-t001:** Routine tasks performed in the construction industry.

Routine Construction Tasks
Clean or prepare the worksite
Load or unload construction materials
Mount of dismount structures and scaffoldings
Dig trenches or prepare the work surfaces
Use work tools
Help fellow workers

**Table 2 ijerph-18-05652-t002:** Demographic characteristics of the sample.

Parameter *n* = 112	Frequency	%	Parameter *n* = 112	Frequency	%
Gender			Age range (years)		
Male	111	99.11	20–25	15	13.39
Female	1	0.89	26–30	19	16.96
			31–35	14	12.50
			36–40	12	10.71
Nationality			41–45	18	16.07
Mexico	63	56.29	46–50	9	8.04
USA	49	43.75	51–55	12	10.71
			>55	13	11.61
Residence in Paso del Norte			Time of residence in the USA		
Yes	70	62.50	Less than one year	4	3.57
No	25	22.32	Between 1 and 3 years	8	7.14
Other	17	15.18	Between 4 and 6 years	9	8.04
			More than six years	91	81.25
Time working in the construction industry			Time worked in current company		
More than 12 months	85	75.89	Less than one year	31	27.68
Less than 12 months	11	9.82	Between 1 and 3 years	23	20.54
Other	16	14.29	Between 4 and 6 years	19	16.96
			More than six years	39	34.82

**Table 3 ijerph-18-05652-t003:** Estimation of perceived risk during routine construction tasks.

Perception of:						
Routine Construction Tasks	Injury Probability		Injury Severity		Risk	
(*n* = 112)	Mean	S.D.	Mean	S.D.	Mean	S.D.
Clean or prepare your workplace.	3.7589	1.36392	3.7297	1.38134	15.27	8.811
Load or unload construction materials.	3.5536	1.25805	3.4464	1.3345	13.32	7.972
Mount or dismount structures and scaffoldings.	2.5804	1.37309	2.5446	1.31443	7.89	7.042
Dig trenches or prepare work surface.	2.9375	1.42235	2.9018	1.4763	9.86	7.908
Use work tools.	3.9375	1.21018	3.8304	1.2797	15.9	7.951
Help fellow workers.	4.0982	1.11468	3.875	1.37628	16.7	8.261

**Table 4 ijerph-18-05652-t004:** Factorial analysis.

Kaiser–Meyer–Olkin Measure of Sampling Adequacy		0.784
Bartlett’s Test of Sphericity	Approximate Chi-Square	435.086
	df	15
	Significance	0.000

df = Degrees of freedom.

**Table 5 ijerph-18-05652-t005:** ANOVA tables for risk perception.

	Sum of Squares	dF	Mean Square	F	Sig.
Age					
Between Groups	3.007	7	0.430	0.418	0.890
Within Groups	107.012	104	1.029		
Total	110.019	111			
Time of residence in the USA					
Between Groups	10.506	3	3.502	3.801	0.012
Within Groups	99.513	108	0.921		
Total	110.019	111			
Time performing construction work					
Between Groups	5.480	3	1.827	1.887	0.136
Within Groups	104.539	108	0.968		
Total	110.019	111			

**Table 6 ijerph-18-05652-t006:** Tukey test for risk perception dependent on time of residence in the United States.

(I) Time of Residence in the USA	(J) Time of Residence in the USA	Mean Difference (I–J)	Std. Error	Sig.	95% Confidence Interval	
					Lower Bound	Upper Bound
1	2	−1.13601222	0.58781802	0.221	−2.6699143	0.3978899
	3	−0.07344599	0.57682980	0.999	−1.5786745	1.4317825
	4	−0.98682157	0.49038635	0.190	−2.2664772	0.2928341
2	1	1.13601222	0.58781802	0.221	−0.3978899	2.6699143
	3	1.06256623	0.46642893	0.110	−0.1545729	2.2797054
	4	0.14919065	0.35398036	0.975	−0.7745157	1.0728970
3	1	0.07344599	0.57682980	0.999	−1.4317825	1.5786745
	2	−1.06256623	0.46642893	0.110	−2.2797054	0.1545729
	4	−0.91337557	0.33541719	0.037	−1.7886416	−0.0381096
4	1	0.98682157	0.49038635	0.190	−0.2928341	2.2664772
	2	−0.14919065	0.35398036	0.975	−1.0728970	0.7745157
	3	0.91337557	0.33541719	0.037	0.0381096	1.7886416

## Data Availability

The data presented in this study are available on request from the corresponding author. The data are not publicly available due to protection of participants.

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
