# Peer review of "Differences in Workplace Risk Perception between Foreign-Born and First-Generation Mexican American Construction Workers"

_ijerph, 2021, doi:10.3390/ijerph18115652_

Round 1
Reviewer 1 Report
This study aims to determine significant differences in the risk perception between Mexican American migrants and first-generation Mexican American construction workers. However, its quality should be improved by the following comments.
For the introduction part,
- The latest accident statistics of the Mexican American industry should be provided.
- The definition of risk perception is missing. The authors can obtain the definition from previous studies such as “Risk-taking behaviors of Hong Kong construction workers–A thematic study”, “Risk perception and risk-taking behavior of construction site dumper drivers”, etc.
- Risk perception not only affects person’s health behaviors but also the risk-taking behavior and the use of personal protective equipment of construction workers (in Hong Kong and other regions). Please add some recent work to demonstrate this point.
For the methodology part,
- the sample size of this study is relatively small. Are the data sufficient to obtain robust results? Please explain.
- There is a valid and reliable scale called the construction worker risk perception (CoWoRP) scale for measuring the risk perception of construction workers in the literature. Also, the CoWoRP scale considers cognitive (probability and severity) and affective risk perception (worry and unsafe) of construction workers. In this study, only cognitive risk perception of construction workers was considered. Why did the authors not use the CoWoRP scale? I think the tasks in Table 1 cannot sufficiently represent the hazardous situations for participants to perceive risk and the affective risk perception is also important to measure. The authors should think about the use of the CoWoRP scale. This is a limitation of study which should be clearly stated in the discussion and future studies can use the CoWoRP scale to compare the risk perception between Foreign-born and First-generation Mexican American Construction Workers.
- Why was there only one female participant in this study?
- The reliability of the measurement should be tested with Cronbach’s alpha if possible.
For discussion part
- Apart from cultural difference, there are many factors that influence risk perception of construction workers. The authors should add more discussion on this topic. For example, it was found that safety training can positively influence affective risk perception but negatively correlated with cognitive risk perception in the work “The effect of personal and organizational factors on the risk-taking behavior of Hong Kong construction workers”.
- The practical recommendations should be provided in a separate paragraph in discussion.
Reviewer 2 Report
The Abstract should be improved, as it would be better for the article if the authors put here some of the values obtained throughout the research. It is important to captivate the reader of the article at this early stage.
Why did the authors in the Literature Review only focus on construction in the United States of America, and not extend the search worldwide? It would have been better for the quality of the research, and to be able to compare it with other countries and other cultures. This would have improved the scientific contribution of this article to the scientific community.
[line 60-61] The authors state that “… construction workers in the United States have an overall 75% probability of undergoing a disabling injury …”. How was the percentage calculated? Where are the sources of this information? This is something very relevant, so the authors should be careful when making this statement.
[line 79] The authors state that “In construction, perceived risk is widely used to quantify risks in this industry.” But isn't risk perception widely used, by any kind of industry and service, to make workers aware of compliance with safety standards? Only in construction? I have big doubts!
[line 130] In “Table 1" is very incomplete and there are several specialty activities that normally occur in the works that have not been considered by the authors, and which would have made a greater contribution to the scientific community. For example, why was welding, which has a high fire risk, not considered? Why wasn't the painting of buildings, where there is direct contact with chemical substances, considered? Among other activities...
[2.3 Procedure] How was the questionnaire initially validated? If the questionnaire was designed by the authors, and not validated, having been applied follow-up to workers, there are many doubts about the conclusions of this research...
How did the authors study the reliability of the questionnaire applied? Did the authors calculate Cronbach's Alpha? It is not clear how the authors made the study of the reliability of the responses obtained from the questionnaire?
Round 2
Reviewer 1 Report
The authors did a good job in addressing my comments. I have no further comments.
Reviewer 2 Report
The article has been substantially improved, which helps in the understanding of the published research. Congratulations to the authors for their excellent work!